# Differential Expression of Two Copies of the *irmA* Gene in the Enteroaggregative *E. coli* Strain 042

M. Bernabeu,[a] S. Aznar,[b] A. Prieto,[a] M. Hüttener,[a] A. Juárez[a,b]

[a]Department of Genetics, Microbiology and Statistics, Universitat de Barcelona, Barcelona, Spain
[b]Institute for Bioengineering of Catalonia, The Barcelona Institute of Science and Technology, Barcelona, Spain

**ABSTRACT** Gene duplications significantly impact the gene repertoires of both eukaryotic and prokaryotic microorganisms. The genomes of pathogenic *Escherichia coli* strains share a group of duplicated genes whose function is mostly unknown. The *irmA* gene is one of the duplicates encoded in several pathogenic *E. coli* strains. The function of its gene product was investigated in the uropathogenic *E. coli* strain CFT073, which contains a single functional copy. The IrmA protein structure mimics that of human interleukin receptors and likely plays a role during infection. The enteroaggregative *E. coli* strain 042 contains two functional copies of the *irmA* gene. In the present work, we investigated their biological roles. The *irmA_4509* allele is expressed under several growth conditions. Its expression is modulated by the global regulators OxyR and Hha, with optimal expression at 37°C and under nutritional stress conditions. Expression of the *irmA_2244* allele can only be detected when the *irmA_4509* allele is knocked out. Differences in the promoter regions of both alleles account for their differential expression. Our results show that under several environmental conditions, the expression of the IrmA protein in strain 042 is dictated by the *irmA_4509* allele. The *irmA_2244* allele appears to play a backup role to ensure IrmA expression when the *irmA_4509* allele loses its function.

**IMPORTANCE** Gene duplications occur in prokaryotic genomes at a detectable frequency. In many instances, the biological function of the duplicates is unknown, and hence, the significance of the presence of multiple copies of these genes remains unclear. In pathogenic *E. coli* isolates, the *irmA* gene can be present either as a single copy or in two or more copies. We focused our work on studying why a different pathogenic *E. coli* strain encodes two functional copies of the *irmA* gene. We show that under several environmental conditions, one of the alleles dictates IrmA expression, and the second remains silent. The latter allele is only expressed when the former is silenced. The presence of more than one functional copy of the *irmA* gene in some pathogenic *E. coli* strains can result in sufficient expression of this virulence factor during the infection process.

**KEYWORDS** enteroaggregative *E. coli*, 042, gene duplications, *aec69*, *irmA*

Gene duplications occur in both eukaryotes and prokaryotes and play a key role in the generation of the functional diversity of genes (1–6). In bacteria, gene duplications have been associated with the adaptation of cells to a changing environment (7, 8) and have been found to occur more frequently among transferred genes than among indigenous genes (9). The incidence of gene duplications can be different across different bacterial genera and species. These duplications can also be strain-specific, species-specific, and genus-specific (10, 11). In *E. coli*, a group of duplicated ORFs is common to almost all the currently sequenced pathogenic strains (10). Several of these ORFs code for proteins of unknown function. One of the duplicated genes in several *E. coli* strains is the *aec69* gene. In a recent work, the *aec69* gene and its gene

Address correspondence to A. Juárez, ajuarez@ub.edu, or M. Hüttener, mhuttener@me.com.

The authors declare no conflict of interest.

product were studied in the uropathogenic *E. coli* strain CFT073 (12). The *aec69* gene was renamed *irmA* in the latter work. The IrmA protein forms a dimer whose structure is similar to those of human cytokine receptors such as the interleukin-2 or the interleukin-4 receptors. In addition, purified IrmA can bind to their cognate cytokines. IrmA likely plays a role during the infection process of the *E. coli* strains that express it. Most of these strains belong to the uropathogenic (UPEC), enterohemorrhagic (EHEC) and enteroaggregative (EAEC) pathotypes (12). The *irmA* gene maps close to the *flu* gene, which codes for the Ag43 outer membrane protein (13). Previous reports have shown that *flu* can also be present in multiple copies (14, 15). In some *E. coli* strains, such as EC958 (which belongs to the ST131 clonal type), the *flu yeeR irmA* gene cluster is present only in a single copy. In other strains, such as the UPEC strain CFT073, it is present in two copies.

Based on the amplification of the *flu yeeR irmA* transcripts of strains EC958 and CFT073 by RT–PCR, as well as the determination of the 5′ transcriptional start points by RACE, it was concluded that *irmA* forms part of a single *flu yeeR irmA* transcriptional unit. In the CFT073 strain, transcription of one of the *flu yeeR irmA* operons is disrupted because of the presence of insertion sequences between *flu* and *yeeR* genes (12). Previous studies have shown that the expression of Ag43 varies across phases and is controlled by an epigenetic mechanism involving the global regulatory protein OxyR and the Dam methyltransferase (16); additionally, it was shown that *irmA* expression is subjected to OxyR repression (12).

The *E. coli* strain 042 is the EAEC prototype. It caused diarrhea in a volunteer trial (17). The genome sequence of this strain is available (18), and its virulence factors have been characterized. The 042 strain harbors the IncFIIA virulence plasmid pAA2 (18, 19), which encodes the virulence master regulator AggR and other virulence determinants, such as the aggregative adherence fimbriae AAF/II (20, 21). When analyzing the 042 genomic sequence, we noticed that it contains several genes that are present in two or more copies, including *irmA* (10). Unlike strain CFT073, no insertion sequences disrupt the *flu yeeR irmA* region in strain 042, and only minor nucleotide changes can be detected. In this work, we aimed to gain insight into the biological role of both *irmA* alleles. We also show that these alleles are differentially expressed. One allele remains silent when the other is expressed. We show that, unlike strain CFT073, transcription of the *irmA* alleles in the 042 strain is dependent upon a promoter that maps to the intergenic *yeeR irmA* region. The biological reason for the existence of duplicates of the *irmA* genes in some pathogenic *E. coli* strains is discussed.

## RESULTS

**Analysis of IrmA protein expression dictated by the *irmA_2244* and *irmA_4509* alleles in the *E. coli* strain 042.** To detect IrmA expression, we first purified the IrmA protein (Fig. S1) and obtained polyclonal antibodies. Expression of the IrmA protein was tested in the culture supernatants of the 042 wt strain and its single *oxyR* and double *irmA_2244 irmA_4509* mutant derivatives (Fig. 1). In accordance with the data previously obtained for *E. coli* strains CFT703 and EC958 (12), IrmA expression was barely detectable in the 042 wt strain, but its expression was significantly increased in the *oxyR* mutant derivative (Fig. 1A).

To improve the detection of the IrmA protein and to assess the differential expression of the gene product controlled by the *irmA_2244* or *irmA_4509* allele, we inserted a Flag-tag downstream of the *irmA_2244* and *irmA_4509* genes. We also constructed *irmA::lacZ* transcriptional fusions with both *irmA* alleles. IrmA expression was then assessed in the 042 ΔLC (042 derivative lacking the *lacZ* and *cat* genes) strain grown in LB medium at 37°C, both by immunodetecting the IrmA protein with anti-Flag monoclonal antibodies and by analyzing the transcription of both *irmA* alleles of strain 042 ΔLC (Fig. 1B and C). The results showed that although one of the *irmA* alleles (*irmA_4509*) is expressed in the exponential growth phase and at the onset of the stationary phase in cultures grown in LB medium at 37°C, the other (*irmA_2244*) is not expressed.

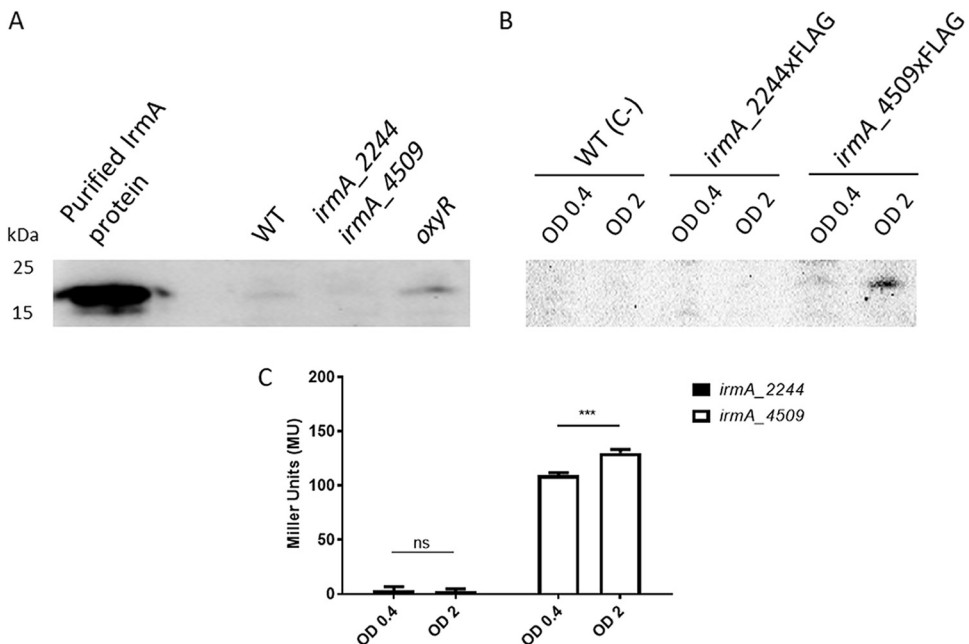

**FIG 1** Immunodetection of the IrmA protein and expression analysis of the *irmA_2244* and *irmA_4509* alleles. (A) Western blot analysis of protein extracts from the 042 wt strain, its double *irmA* derivative and its *oxyR* derivative. Purified IrmA protein was used as a control. (B) Immunodetection of the IrmA protein by using anti-Flag antibodies in samples from cultures grown to an OD of 0.4 and 2.0. Strains 042 wt (used as a negative control), 042 irmA_2244xFLAG and 042 irmA_4509xFLAG were used for the assay. (C) Transcription of the *irmA_2244* and *irmA_4509* alleles, measured as the $\beta$-galactosidase activity of the *irmA_2244::lacZ* and *irmA_4509::lacZ* transcriptional fusions in samples collected at the exponential and early stationary phases of growth (OD$_{600}$ 0.4 and 2.0, respectively). The values obtained are based on triplicate experiments. ns, not significant; ***, $P < 0.001$.

**Expression of the *irmA_2244* allele is silent under several *in vitro* growth conditions.** We next investigated whether expression of the *irmA_2244* allele could be detected when strain 042 was grown outside normal conditions (LB medium at 37°C). First, we analyzed the effect of the growth temperature and the culture medium (rich and minimal). With respect to the *irmA_4509* allele, its expression is increased at high temperature (37°C) and when cells were grown in minimal medium (Fig. 2A to D). Again, no expression was detected for the *irmA_2244* allele.

The next environmental factor studied was the presence of oxygen in the medium. Samples from both aerated cultures and anaerobic cultures were used for IrmA immunodetection and for the analysis of the transcription of the corresponding *irmA::lacZ* fusion. For the *irmA_4509* allele, anaerobic conditions resulted in reduced expression of the gene. Again, expression of the *irmA_2244* allele could not be detected (Fig. 2E and F).

We also analyzed whether nutritional stress could influence the expression of the *irmA* alleles in strain 042. First, we obtained the growth curves of strain 042 ΔLC in minimal M9 medium with different glucose concentrations (Fig. 3). With a glucose concentration of 0.5 g/l, growth ceased at the mid-exponential growth phase. We used cultures grown with either 4 or 0.5 g/l glucose to compare the expression of both *irmA* alleles in strain 042 ΔLC grown with excess glucose or under glucose-limited conditions. With respect to the *irmA_4509* allele, exhaustion of the carbon source resulted in a significant increase in IrmA expression. Again, no expression of the *irmA_2244* allele could be detected (Fig. 3B and C).

**Regulation of the expression of the *irmA* alleles by the H-NS/Hha and OxyR systems.** One of the genes that is also duplicated in strain 042 is the *hha* gene. Its gene product (the Hha protein) interacts with the well-characterized nucleoid-associated protein H-NS to modulate gene expression (22). We found a strong correlation between the presence of additional copies of the *hha* gene (i.e., the *hha2* gene, locus

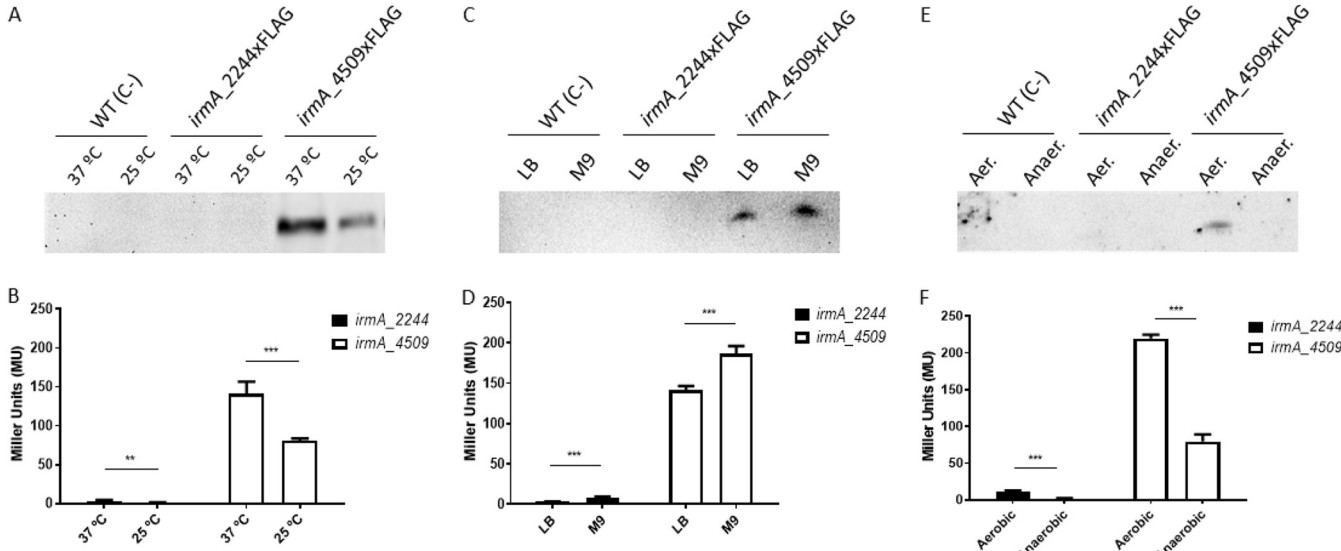

**FIG 2** Effect of different environmental conditions on the expression of the *irmA_2244* and *irmA_4509* alleles. Expression was measured either by immunodetecting the IrmA protein with anti-FLAG monoclonal antibodies (A, C, E) or by measuring the $\beta$-galactosidase activity of the corresponding *irmA::lacZ* transcriptional fusion (B, D, F). The values obtained are based on triplicate experiments. The effect of the growth temperature (A, B), culture medium (C, D) and presence of oxygen (E, F) on the expression of both *irmA* alleles is shown. Aer., aerobiosis; Anaer., anaerobiosis. **, $P < 0.01$; ***, $P < 0.001$.

tag ec042_4516, located upstream of the *irmA_4509* gene) and the *yeeR irmA* cluster in several pathogenic *E. coli* strains, including strain 042 (10). This result suggests that the *irmA* genes of strain 042 can be regulated by Hha (and/or its HhaII allele) and by H-NS. To confirm this result, we constructed the corresponding *hns*, *hha* and *hha hha2* mutant derivatives of strains 042 *irmA_2244xFLAG* and 042 *irmA_4509xFLAG* and analyzed IrmA expression in the different strains. With respect to the expression of the *irmA_4509* allele, its expression was increased in the double *hha hha2* mutant but not in the *hns* mutant (Fig. 4A and B). For the *irmA_2244* allele, no effect was found (Fig. 4A).

We also decided to analyze whether the previously reported repression of the *irmA* genes by OxyR (12) responds to oxidative stress. We established optimal conditions to generate oxidative stress in the 042 strain (see Materials and Methods section for details) and analyzed the effect of oxidative stress on the expression of both *irmA* alleles (Fig. 4C). No expression of the *irmA_2244* allele could be detected under oxidative stress conditions or in the *oxyR* mutant derivative of strain 042. In contrast, the expression of the *irmA_4509* allele was significantly increased in the *oxyR* mutant derivative, but the effect was independent of oxidative stress.

**Expression of the *irmA_2244* allele can be detected in an *irmA_4509* mutant derivative of strain 042.** Under all of the above reported conditions, we were unable to detect the expression of the *irmA_2244* allele. We then decided to test the individual expression of each of the alleles in the absence of the other. Expression was monitored by immunodetecting the IrmA protein produced by the corresponding *irmA* allele with polyclonal anti-IrmA antibodies and by measuring the transcriptional activity of the already obtained *irmA::lacZ* fusions.

The expression of the *irmA_4509* allele in the 042 Δ*irmA_2244* strain was similar to that measured in the 042 wt strain (Fig. 5A). Expression of the *irmA_2244* allele in the 042 Δ*irmA_4509* strain could barely be detected by Western blotting but not at the transcriptional level. When the *oxyR* allele was introduced into the 042 Δ*irmA_2244* and 042 Δ*irmA_4509* strains, this resulted, as expected, in increased expression of IrmA. In this genetic background, the expression of the 042 *irmA_2244* allele could be detected both by Western blotting and by measuring the transcription of the *irmA_2244::lacZ* fusion (Fig. 5A to D). These results were corroborated by detecting the expression of the Flag-tagged alleles in the corresponding mutants by using anti-Flag monoclonal antibodies (Fig. 5B–C).

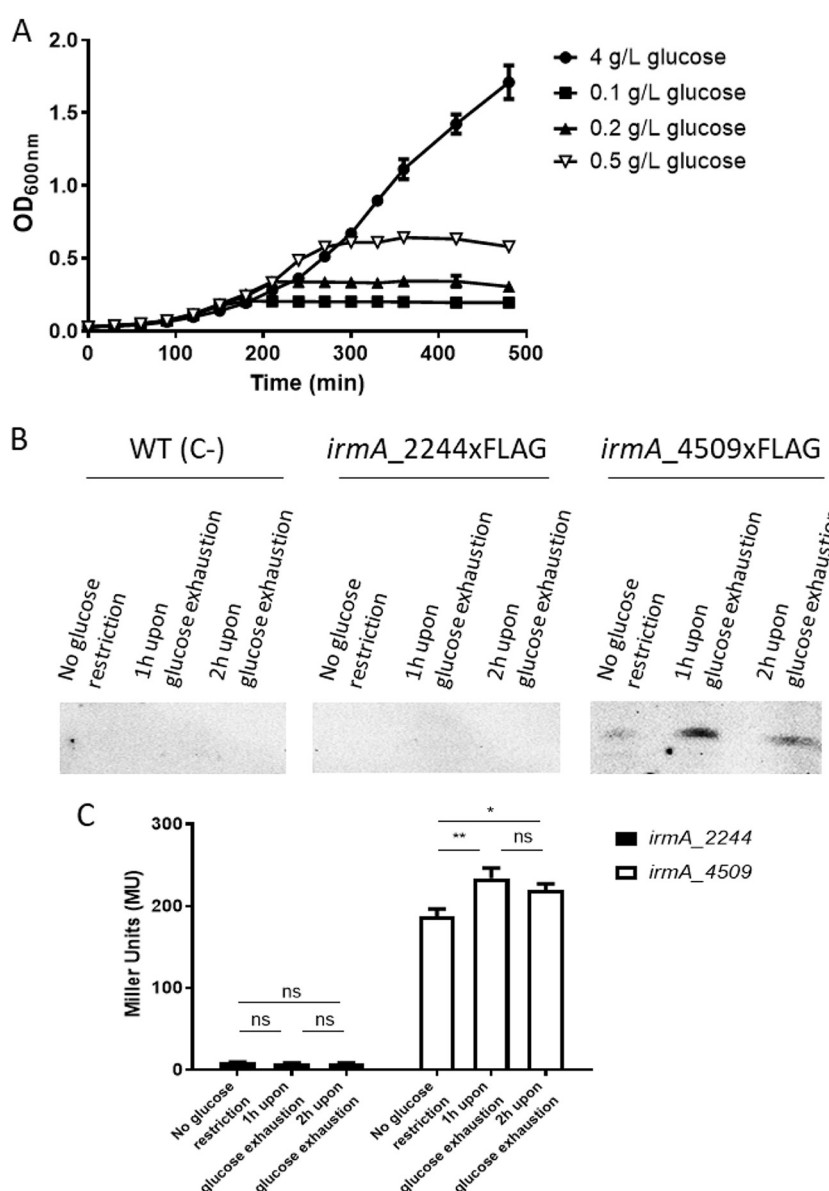

**FIG 3** Effect of nutritional stress on the expression of the *irmA_2244* and *irmA_4509* alleles. (A) Growth curves of strain 042 ΔLC in M9 minimal medium with different glucose concentrations. (B) Immunodetection of the IrmA_2244xFLAG and IrmA_4509xFLAG proteins with anti-FLAG monoclonal antibodies. (C) β-galactosidase activity of the strains harboring the *irmA_2244::lacZ* and *irmA_4509::lacZ* transcriptional fusions. The values obtained are based on triplicate experiments. ns, not significant; *, $P <$ 0.05; **, $P <$ 0.01.

***irmA* transcripts are initiated downstream of the *yeeR* gene in strain 042.** The results reported above show that under a wide range of conditions, the expression of the *irmA_2244::lacZ* allele is significantly lower than that of the *irmA_4509::lacZ* allele. We then compared the corresponding regulatory regions of both genes to identify modifications to the nucleotide sequences that might account for the observed differences in gene expression. Nucleotide differences can be found either in the *flu* regulatory region or in the intergenic *yeeR irmA* region (Fig. S2).

As the *irmA* gene has been reported in strain CFT073 to be cotranscribed with *flu*, we obtained *flu::lacZ* gene fusions by cloning the *flu* promoter corresponding to nucleotides −105 to +22 into the promoterless plasmid pUJ8 and analyzed β-galactosidase expression in four clones harboring each of the constructs. The same expression pattern was obtained for all clones of both constructs (Fig. 6). For each gene fusion construct, two of

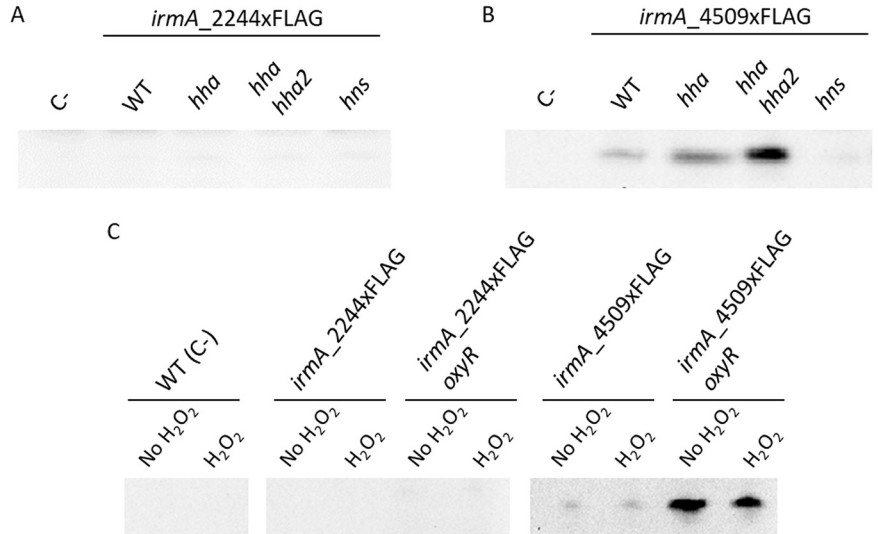

**FIG 4** Effect of the Hha and H-NS proteins and of the oxidative stress on the expression of the *irmA_2244* and *irmA_4509* alleles. Immunodetection of the IrmA_2244xFLAG and IrmA_4509xFLAG proteins with anti-FLAG monoclonal antibodies in cell extracts from the 042 strain harboring either the irmA_2244xFLAG or irmA_4509xFLAG derivatives, alone (wt) or combined with *hha*, *hhahha2* (Hha null mutant) and *hns* derivatives (A and B). (C) Immunodetection of the IrmA_2244xFLAG and IrmA_4509xFLAG proteins with anti-FLAG monoclonal antibodies in cell extracts from the 042 wt strain and its irmA_2244xFLAG, irmA_2244xFLAG *oxyR*, irmA_4509xFLAG and irmA_4509xFLAG *oxyR* derivatives.

four clones showed very low $\beta$-galactosidase activity, and the other two clones showed high $\beta$-galactosidase activity. Considering that *flu* is subjected to phase variation, these results can be interpreted as the different clones expressing the *flu* operon in the on/off phases. As these results did not support the observed differences in the expression of the *irmA_2244* and *irmA_4509* alleles, we searched for other differences in the DNA sequence downstream of the *flu* gene. Some alterations in the nucleotide sequence appeared in the intergenic region between *yeeR* and *irmA*, where a putative promoter sequence was detected bioinformatically (Fig. S3). Hence, we decided to determine the transcriptional start point for both *irmA* alleles in strain 042.

We first analyzed the transcription of the *yeeR irmA* region of strains 042 wt, 042 *irmA_2244* and 042 *irmA_4509* by walking RT–PCR. We used a fixed reverse primer within *irmA* and three forward primers located within *irmA*, in the *yeeR irmA* intergenic region or in the *yeeR* coding sequence. Although amplification products could be obtained for primer pairs located in the *irmA-irmA* and *irmA*-intergenic regions, no amplification product could be obtained when the primer pairs mapping in the *irmA* and *yeeR* regions were used (Fig. 7). These results strongly suggest that both *irmA* alleles are transcribed from their own promoters located downstream of the *yeeR* gene. To confirm this result, we determined the transcriptional start point for both the *irmA_2244* and *irmA_4509* alleles by 5′ RACE (Fig. 8). The transcriptional start point was found to occur downstream of the −10 sequence of the bioinformatically identified promoter (Fig. 11). The distance that exists between the bioinformatically identified −10 sequence and the identified transcriptional start point is too long. Hence, it cannot be ruled out that the true −10 sequence of the *irmA* promoter corresponds to the ACCTTTTA sequence of the *irmA_4509* promoter region.

To further support that *irmA* transcription is initiated at the intergenic *yeeR irmA* region, we mutated the *flu* promoter sequence corresponding to the *flu4511* allele (*p4511*) in strain 042 *irmA_4509*xFLAG and compared the IrmA expression in strains 042 wt, 042 *irmA_4509*xFLAG and 042 *irmA_4509*xFLAG *p4511*. Deletion of the *flu p4511* promoter did not influence IrmA expression (Fig. 9), but significantly reduced *flu* transcription levels (Fig. S4).

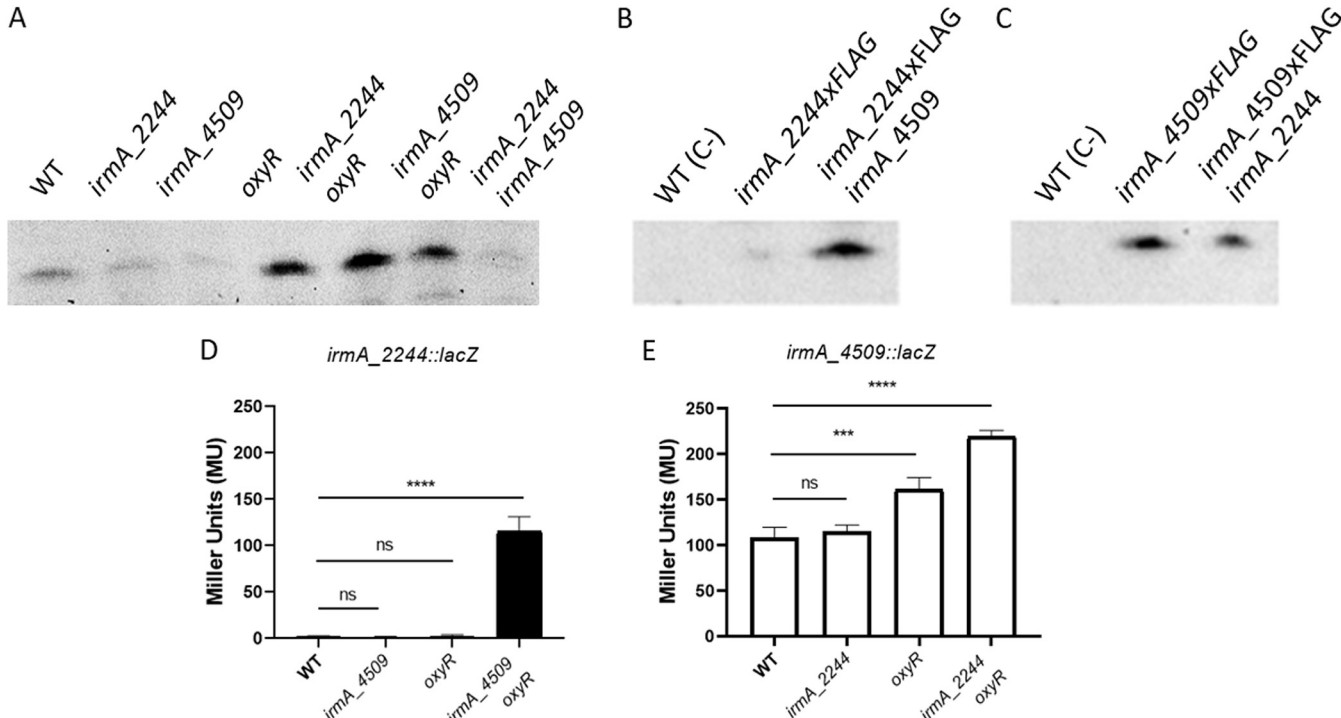

**FIG 5** Expression of each *irmA* allele alone in the 042 strain. (A) Immunodetection of the IrmA_2244 and IrmA_4509 proteins in cell extracts from the wt, *irmA_2244*, and *irmA_4509* strains, as well as their *oxyR* derivatives, using polyclonal anti-IrmA antibodies. (B) Immunodetection of the IrmA_2244xFLAG protein in the wt strain and its irmA_2244xFLAG and irmA_2244xFLAG *irmA_4509* derivatives by using anti-Flag monoclonal antibodies. (C) Immunodetection of the IrmA_4509xFLAG protein in the wt strain and its irmA_4509xFLAG and irmA_4509xFLAG *irmA_2244* derivatives by using anti-Flag monoclonal antibodies. (D, E) *β*-galactosidase activity of the strains harboring the *irmA_2244::lacZ* and *irmA_4509::lacZ* transcriptional fusions in the 042 ΔLC (indicated as wt), *irmA_2244*, and *irmA_4509* strains, as well as their *oxyR* derivatives. The values obtained are based on triplicate experiments. ns, not significant; ***, $P < 0.001$; ****, $P < 0.0001$.

**Alterations in the *irmA* promoter regions account for the differential expression of the *irmA_2244::lacZ* and *irmA_4509::lacZ* alleles.** As detailed above, there are different nucleotide changes in the intergenic region between *yeeR* and *irmA* (Fig. 8), with some mapping in the bioinformatically identified *irmA* promoter. We hypothesized that these changes are responsible for the reduced expression of the *irmA_2244* allele. To support this assumption, we cloned the intergenic region of both the *irmA_2244* and *irmA_4509* alleles (Fig. 10A) in the promoterless vector pUJ8. The constructs were transformed into strain 042 ΔLC, and *β*-galactosidase activity was determined for both constructs (Fig. 10B). In accordance with the previous data obtained, the construct harboring the intergenic region of the *yeeR-irmA_4509* allele showed much higher *β*-galactosidase activity than the construct that incorporated the intergenic region corresponding to the *irmA_2244* allele (Fig. 10B). To demonstrate that nucleotide changes in the promoter region account for the observed reduced expression of the *irmA_2244* allele, we used a site-directed mutagenesis approach to modify the *yeeR-irmA_4509* intergenic region, specifically nucleotides +1, −10, −19 and −20 (Fig. 11A), thus partially transforming the *yeeR-irmA_4509* intergenic region into the *yeeR-irmA_2244* intergenic region. Strain 042 ΔLC harboring plasmid pUJ8-mut*prom4509 was used to detect *irmA* transcription, and strain 042 irmA_4509xFLAG p4509 (lacking the intergenic region between *yeeR* and *irmA*) was used to detect IrmA expression by Western blotting (Fig. 11B and C). Individual point mutations did not have a significant effect on *irmA* expression (data not shown), but the combined alterations in the 042 *irmA_4509* promoter region resulted in reduced expression of IrmA.

**The *flu yeeR irmA* cluster in pathogenic *E. coli* strains.** We also bioinformatically analyzed the *flu yeeR irmA* gene cluster in *E. coli* strains belonging to different pathotypes. Duplications, deletions, and insertions between *flu* and *yeeR* occur in several strains (Fig. 12A). Of the 10 strains analyzed, 4 harbored duplications of the *flu yeeR*

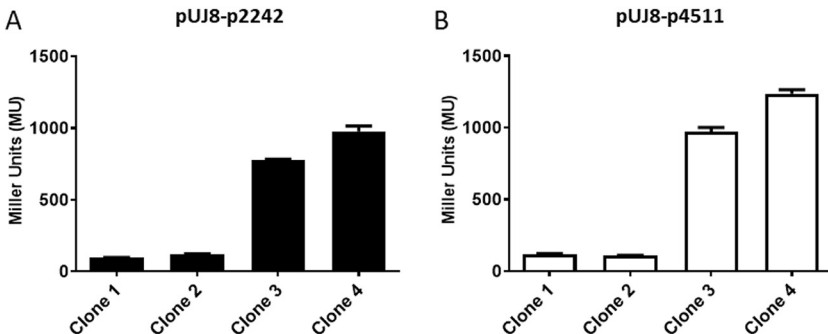

**FIG 6** β-galactosidase activity of plasmidic transcriptional fusions with the *flu* promoter. (A) *flup2242* fusion. (B) *flup4511* fusion. The values obtained are based on triplicate experiments with four independent transformants for each of the plasmids.

*irmA* gene cluster. We also performed a phylogenetic analysis of the intergenic *yeeR-irmA* and the nucleotide sequence of the *irmA* gene (Fig. 12B). The *irmA_4509* allele shows close proximity to the corresponding *irmA* alleles of the EAEC strains 55989 and 0104:H4 and shows a large distance from the *irmA_2244* allele.

## DISCUSSION

When gene duplications occur, their fate (fixation or elimination) may depend on the occurrence of positive selection (23). Different hypotheses have been proposed to

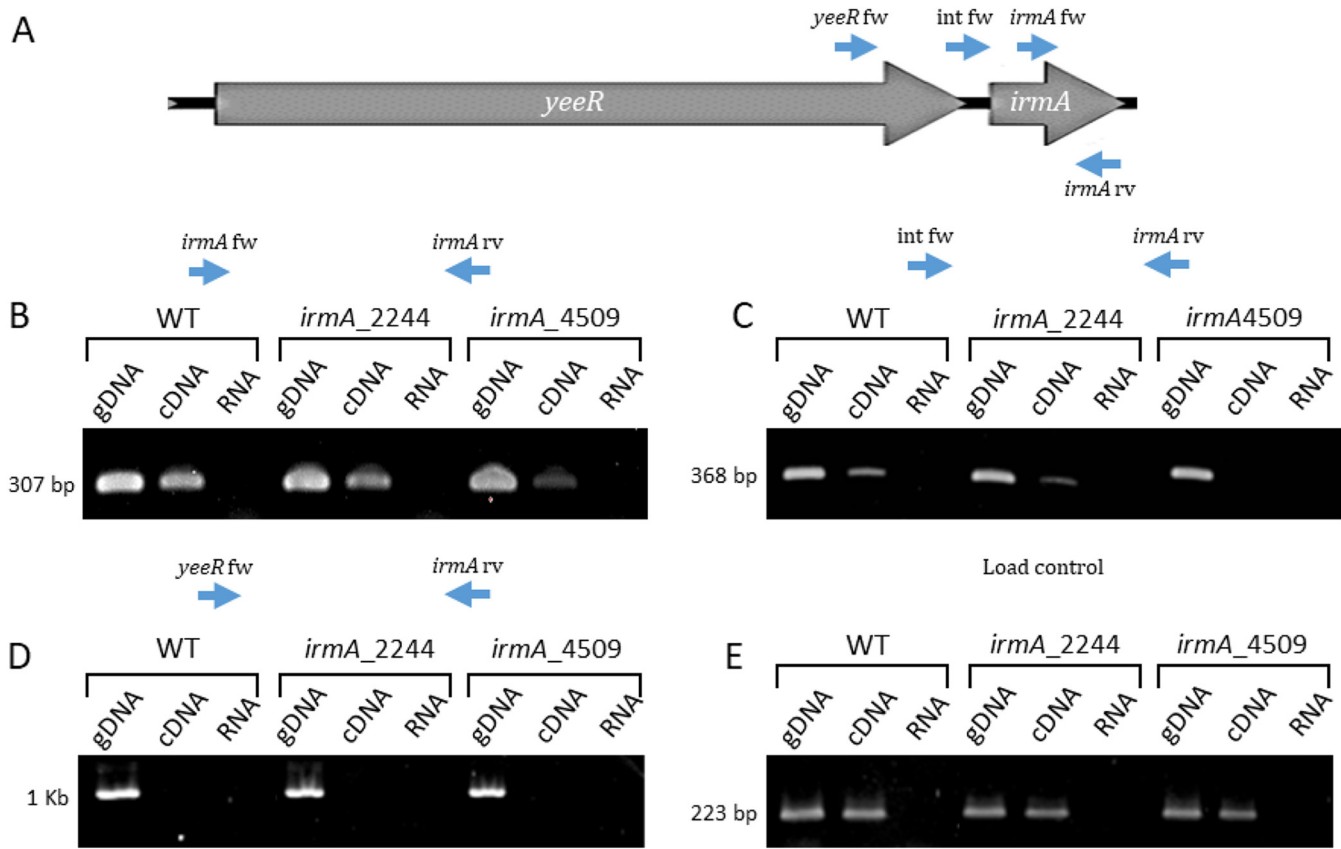

**FIG 7** Analysis of the transcription of the *irmA* operon by walking RT–PCR. (A) Diagram of the oligonucleotides used in the experiment. (B) Amplification products obtained with the oligonucleotide pair *irmA* fw-*irmA* rv. (C) Amplification products obtained with the oligonucleotide pair Int fw-*irmA* rv. (D) Amplification products obtained with the oligonucleotide pair *yeeR* fw-*irmA* rv. (E) Load control amplifying the *gapA* gene (*gapA* fw-*gapA* rv). The expected size of the amplicons is indicated. gDNA: genomic DNA, cDNA: complementary DNA. The 042 wt, *irmA_2244* and *irmA_4509* strains were used for the analysis.

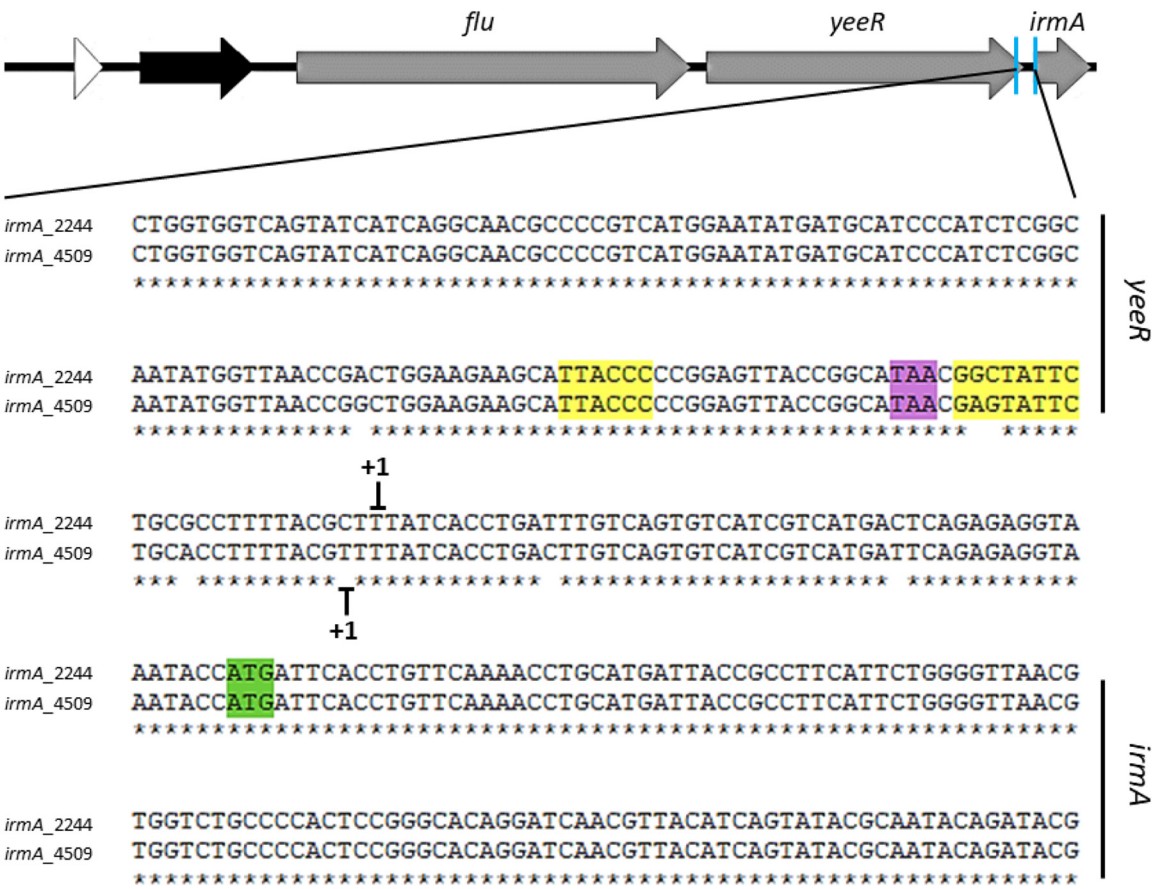

**FIG 8** Identification of the *irmA* promoter by BPROM software and the transcriptional start point of the *irmA* alleles by 5′ RACE. The transcriptional start point identified by 5′ RACE is marked with +1, and the putative -10 and −35 boxes found by using BPROM software are highlighted in yellow. The stop codon of the *yeeR* gene is colored purple, and the start codon of the *irmA* gene is labeled green.

justify the selective advantage of duplicates. First, duplication results in a higher expression level, which can be favorable (24). In this case, expression of the ancestral system should be far from optimal, and increasing the expression level should be beneficial. This can occur in extreme environmental circumstances but not in stable environments to which the organism should be adapted (25). Therefore, it is not surprising that several of the reported examples of gene duplications in bacteria correspond to those living in stressful environments (24, 26–28).

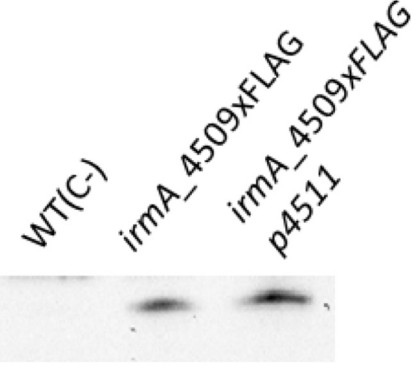

**FIG 9** Influence of deletion of the *flu_4511* promoter (*p4511*) on the expression of the *irmA_4509* gene. Immunodetection of IrmA_4509xFLAG in the 042 wt strain and its irmA_4509xFLAG Δp4511 derivatives. Protein extract from the 042 wt strain was used as a negative control.

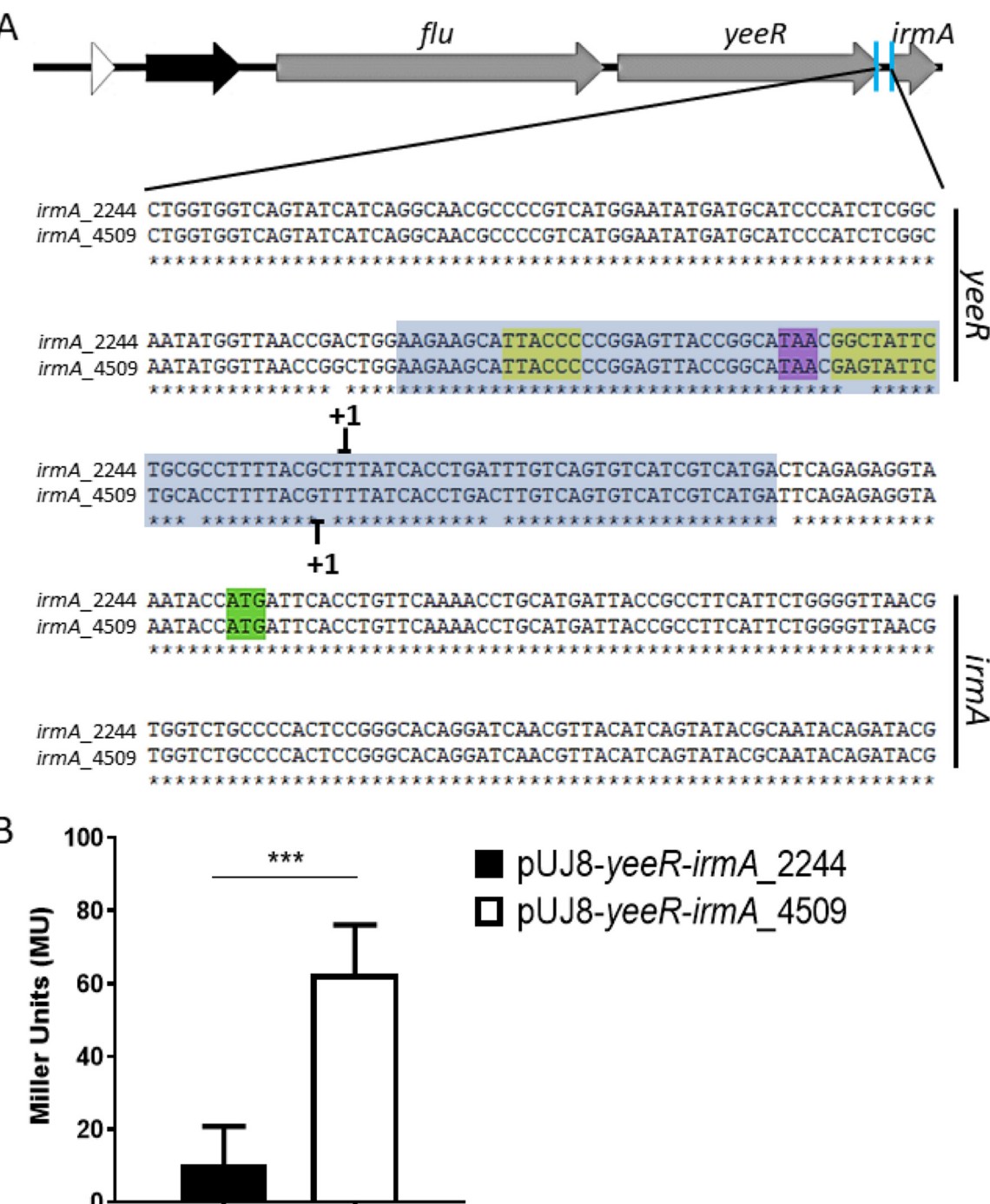

**FIG 10** Transcriptional activity of the intergenic region between the *yeeR* and *irmA* genes. (A) Intergenic *yeeR*/*irmA* region. The cloned sequence is marked with a blue box. The transcriptional start points of the *irmA* genes identified by 5′ RACE are marked with +1. The putative -10 and −35 boxes found by using BPROM software are highlighted in yellow. The stop codon of the *yeeR* gene is colored purple, and the start codon of the *irmA* gene is colored green. (B) β−galactosidase activity of plasmid transcriptional fusions with the intergenic *yeeR-irmA* region of both *irmA* alleles. The values obtained are based on triplicate experiments. ***, $P < 0.001$.

Second, the presence of the second copy of a gene may allow the accumulation of beneficial mutations (29, 30). In addition to these two situations, duplications can also be positively selected when errors in gene expression occur (31). Errors can be either of genotypic or phenotypic origin. Errors in phenotype may be due to the existence of stochastic fluctuations in gene expression (32). The accuracy in the response of noisy genes can be improved by their corresponding duplicates (33–35). Duplicates can also behave as molecular backups for their paralogs (36).

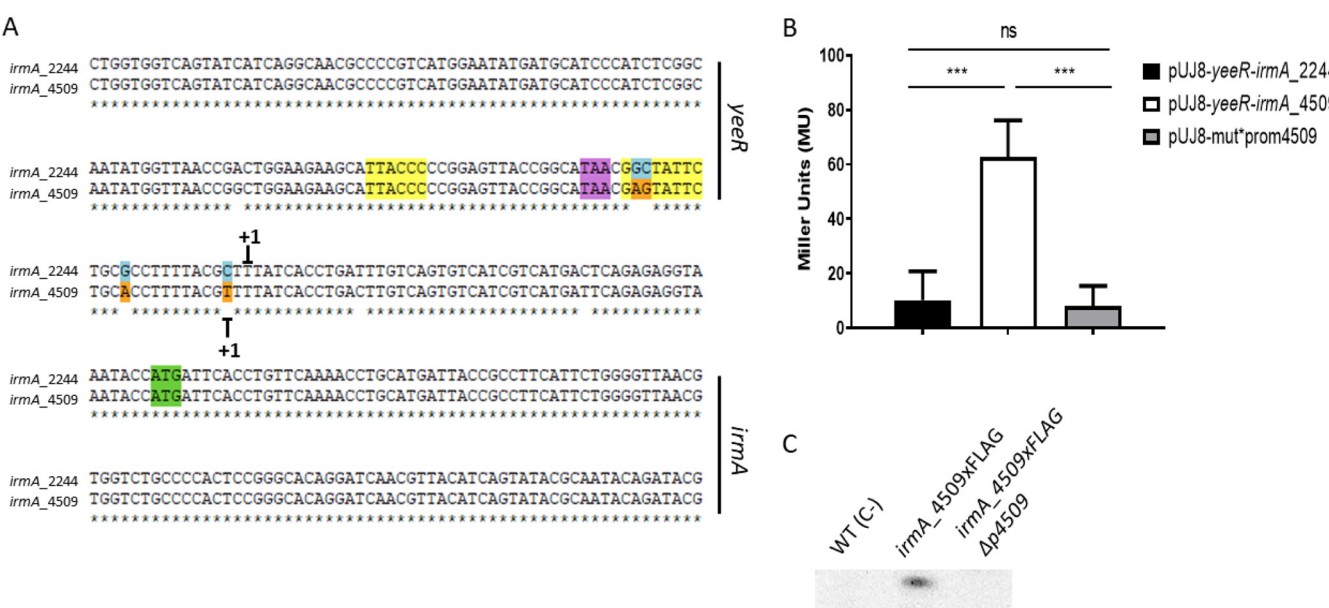

**FIG 11** Differences in the intergenic region between the *yeeR* and *irmA* genes account for the differential expression of both *irmA* alleles. (A) Intergenic *yeeR/irmA* region. Selected point mutations that have been introduced into the sequence are colored blue and orange. The transcriptional start points of the *irmA* alleles identified by 5′ RACE are marked with +1, and the putative −10 and −35 boxes found by using BPROM software and are highlighted in yellow. The stop codon of *yeeR* is colored purple, and the start codon of *irmA* is colored green. (B) β-galactosidase activity of plasmid transcriptional fusions performed with the wt *irmA_2244*, wt *irmA_4509* and mutant *irmA_4509* promoters. (C) Immunodetection of the IrmA_4509xFLAG protein in the wt strain and its irmA_4509xFLAG and irmA_4509xFLAG *p4509* derivatives. A protein extract from strain 042 wt was used as a negative control. The values obtained are based on triplicate experiments. ***, $P < 0.001$.

The *flu yeeR irmA* cluster shows high genetic variability in *Escherichia coli* strains. It is widely distributed among the different pathotypes. Even for a single pathotype such as UPEC, the cluster can be present either as a single copy or as a duplicate (12), and insertion elements may disrupt the structure of one of the copies, as is the case for strain CFT073. We show in this work that IS-mediated disruptions of *flu yeeR irmA* are frequent among strains belonging to different pathotypes. Insertions and deletions occur mainly within the *flu yeeR* region. Nevertheless, other strains, such as EAEC strain 042, contain two copies of the cluster. These copies map in a large duplicated region in the EAEC strain 042. In virulent *E. coli* isolates, the presence or absence of specific genetic determinants, which can also be present in one or more copies, argues for their corresponding gene products being expressed only under very specific infection conditions, which in turn supports the sophisticated evolution of pathogenic *E. coli*.

In contrast to the results obtained for the *flu yeeR irmA* determinant of the UPEC strains EC958 and CFT073, transcription of the *irmA_4509* allele occurs independently of *flu* and is tightly regulated by several factors. As described for the *irmA* gene of strains EC958 and CFT073 (12), transcription of the *irmA_4509* allele of the 042 strain is repressed by OxyR, but OxyR-mediated repression does not appear to respond to the occurrence of oxidative stress conditions. Environmental factors reported to influence the expression of virulence factors also influence the expression of the *irmA_4509* gene (i.e., growth temperature, culture medium). The stress generated by the exhaustion of the carbon source also induces *irmA_4509* expression. High temperature, reduced nutrient supply, and nutritional stress are conditions that strain 042 must cope with in the intestine.

With respect to the regulation of the expression of the *irmA_4509* gene by the H-NS/Hha system, we show here that this gene is subjected to Hha-mediated regulation but not to H-NS-mediated regulation. In several examples of different virulence factors, the Hha protein was found to fine tune the H-NS regulatory activity (22, 37–39). Nevertheless, Hha-mediated regulation of gene expression independent of H-NS has been reported previously, but the mechanism by which this happens remains to be elucidated (40).

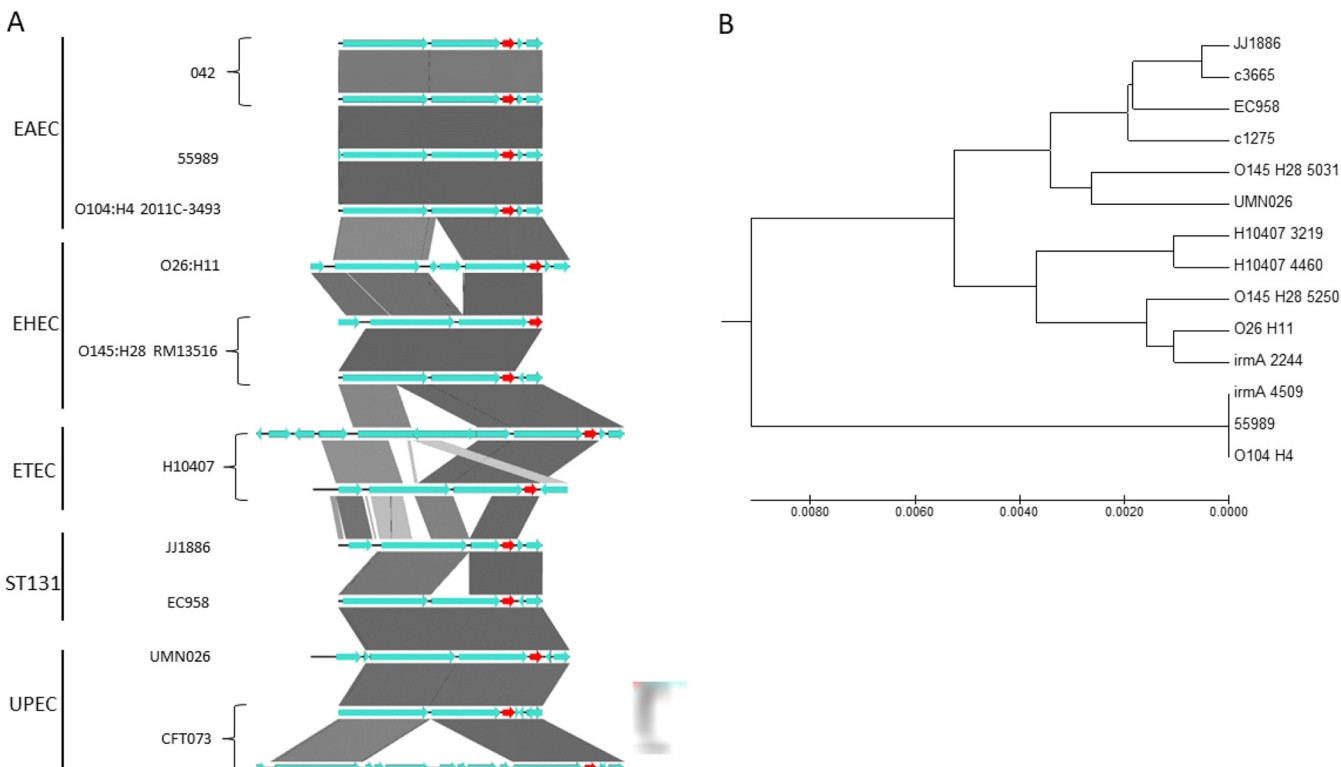

**FIG 12** Bioinformatic analysis of the *flu-yeeR-irmA* region. (A) Alignment of the *flu-yeeR-irmA* region of 10 pathogenic strains of *E. coli* with different pathotypes. The pathotype to which they belong is shown on the left. The *irmA* gene is shown in red. The figure was made with Easyfig. (B) Phylogenetic analysis of the intergenic *yeer-irmA* region and *irmA* gene nucleotide sequence. The evolutionary history was inferred using the UPGMA method. The optimal tree with a total of branch length equal to 0.03874617 is shown. The tree is drawn to scale, with branch lengths in the same units as those of the evolutionary distances used to infer the phylogenetic tree. The evolutionary distances were computed by using the maximum composite likelihood method in units of the number of base substitutions per site.

The *irmA_2244* allele appears to be silent under all the environmental conditions tested that lead to *irmA_4509* expression in strain 042. This phenomenon may be due to the existence of nucleotide substitutions in the *irmA* promoter region. We support this assumption by demonstrating that alterations in the sequence of the *irmA_4509* promoter region that transform it into the *irmA_2244* promoter region abolish *irmA_4509* expression.

In the 042 strain, expression of the *irmA_2244* allele was detected in *irmA_4509* mutants. The mechanism underlying *irmA_2244* induction when *irmA_4509* expression ceases remains to be elucidated. The observed dependence of the expression of one of the duplicated alleles on the expression of the other suggests that the *irmA_2244* allele is a backup for the *irmA_4509* allele in the 042 strain. Several virulent *E. coli* strains encode two copies of the *irmA* gene. In some of these strains, one of the copies has lost its function because of the insertion of foreign DNA sequences. Whether both copies of the *irmA* gene remain functional may rely on the relevance of the IrmA protein itself in the infective process of the corresponding strain. In the EAEC 042 strain, the existence of the *irmA_2244* allele ensures IrmA expression even when the function of the *irmA_4509* allele is lost.

## MATERIALS AND METHODS

**Bacterial strains, plasmids, and growth conditions.** The bacterial strains and plasmids used in this work are listed in Tables S1 and S2. The bacterial strains were routinely grown in Luria-Bertani (LB) medium (10 g l$^{-1}$ NaCl, 10 g l$^{-1}$ tryptone and 5 g l$^{-1}$ yeast extract) or in M9 minimal medium supplemented with glucose at a final concentration of 0.4% with vigorous shaking at 200 rpm (Innova 3100, New Brunswick Scientific). The antibiotics used were chloramphenicol (Cm) (25 $\mu$g mL$^{-1}$), tetracycline (Tc) (15 $\mu$g mL$^{-1}$), carbenicillin (Cb) (100 $\mu$g mL$^{-1}$) and kanamycin (Km) (50 $\mu$g mL$^{-1}$) (Sigma–Aldrich).

**Genetic manipulations.** All enzymes used to perform standard molecular and genetic procedures were used according to the manufacturer's recommendations.

Deletions of the *irmA*, *hns*, *hha*, *oxyR,* and *flu* genes were performed in strain 042 by using the λ Red recombination method, as previously described (41). The antibiotic resistance determinants of the

plasmids pKD3/pKD4 was amplified using the corresponding oligonucleotides (P1/P2 series, see Table S3). The mutants were confirmed by PCR using the corresponding oligonucleotides (P1up/P2down series, see Table S3). When necessary, the antibiotic resistance cassette was eliminated using the FLP/FRT-mediated site-specific recombination method, as previously described (42).

A transcriptional *lacZ* fusion was made with the *irmA_2244* and *irmA_4509* genes of strain 042 ΔLC. The antibiotic resistance determinant from kanamycin was eliminated using an FLP/FRT-mediated site-specific recombination method, as previously described, thus generating strains *irmA_2244* and *irmA_4509*. An FRT-generated site was used to integrate the *lacZ* reporter gene encoded in plasmid pKG136 (43), generating transcriptional *irmA_2244::lacZ and irmA_4509::lacZ* fusions.

Recombinational transfer of the Flag sequence into the *irmA_2244* and *irmA_4509* genes was achieved by following a previously described methodology (44). The template vector encoding the Flag sequence and Km$^r$ cassette used was the pSUB11 plasmid. The primers used for construction of the Flag-tagged derivative were the 3× P1/P2 series (Table S3). The correct insertion of the Flag-tag was confirmed by PCR using oligonucleotides from the P1up/P2down series (see Table S3).

Transcriptional fusions stored in plasmidic vectors were generated to study the regulatory regions of genes of interest using the promoterless plasmid pUJ8. The promoter regions to be studied were amplified by PCR using Phusion Hot Start II DNA polymerase (Thermo Scientific) and the corresponding oligonucleotides (Fw/Rv series, see Table S3). The EcoRI and BamHI restriction sites were added at the 5′ and 3′ ends, respectively. Upon amplification, both the insert and the vector were digested and ligated into the EcoRI/BamHI site. The plasmids generated were Sanger sequenced and termed pUJ8-*yeeR-irmA_2244*, pUJ8-*yeeR-irmA_4509* and pUJ8-mut\*prom*4509*.

**Site-directed mutagenesis.** Point mutations were generated in the transcriptional fusions obtained in the plasmid pUJ8, and oligonucleotides with sizes between 25 and 45 nucleotides were designed. These oligonucleotides contained the desired nucleotide change in the center of their sequence (Fw/Rv series, see Table S3). The amplification reaction was performed with Phusion Hot Start II DNA polymerase (Thermo Scientific) to obtain the desired nucleotide mutation with the following program: 3 min at 98℃ for initial denaturation, followed by denaturation at 98℃ for 30 sec, hybridization at 55℃ for 30 sec and extension at 68℃ for 30 sec/kb. The PCR cycle was repeated 18 times, and a final extension was carried out for 10 min at 68℃. Once the amplification reaction had finished, the PCR product was digested with DpnI, thus eliminating the methylated template plasmid. Next, strain DH5$\alpha$ was transformed by electroporation and seeded on LB plates supplemented with carbenicillin. Transformants were selected and checked for the predicted point mutations upon plasmid extraction and Sanger sequencing with the pUJ8 P1up and lacZR primers (Table S3).

**Oligonucleotides.** The oligonucleotides (from 5′ to 3′) used in this work are listed in Table S3.

**$\beta$-Galactosidase assay.** $\beta$-Galactosidase activity measurements were performed as previously described (45). The values are given as Miller units. Student's *t* test was used to determine statistical significance, and the values were obtained by using GraphPad Prism 8 software. A *P* value of less than 0.05 was considered significant.

**Oxidative stress assay.** The optimal conditions to induce oxidative stress in strain 042 were established by modifying the protocol described in (46). Briefly, 10 mL of an overnight culture was grown in LB medium under shaking at 37℃. The next day, a 1:100 dilution was made in 50 mL of M9 medium supplemented with 4 g/L glucose, and cultures were grown at 37℃ under shaking. Upon reaching an $OD_{600nm}$ of 1, oxidative stress was induced by adding 0.05% $H_2O_2$ and incubating for 2 h at 37℃ under shaking. Then, 1 mL of the culture was centrifuged, the supernatant was removed, and the pellet was resuspended in Laemmli buffer (glycerol 5% vol/vol, $\beta$-mercaptoethanol 2.5%, SDS 1.15% p/v, Tris–HCl 31 mM pH 6.6 and bromophenol blue 0.05%).

**Polyclonal antibody production.** For polyclonal antibody production, we purified the His-tagged IrmA protein. Then, the DNA region containing the complete *irmA* coding sequence was amplified by PCR using the genomic DNA from strain 042 as a template and the primers *irmA* pLATE31CT fw and *irmA* pLATE31CT rv together with Thermo Scientific Phusion Hot Start II High-fidelity DNA polymerase following the manufacturer's recommendations. The DNA was then purified using a Thermo Scientific Gen eJet PCR purification kit and ligated into the pLATE31 vector to add a His-tag to the C-terminal end following the manufacturer ̈s instructions (Thermo Scientific LICator LIC cloning and expression system). The resulting plasmid, termed pLATE31-*irmA*, was Sanger sequenced. BL21DE3 cells were used for heterologous recombinant expression of the IrmA protein. Cells transformed with the pLATE31-*irmA* plasmid were grown in LB medium supplemented with carbenicillin at a final concentration of 100 $\mu$g/mL at 37℃ until an $OD_{600nm}$ of 0.4 was reached. Then, protein expression was induced by adding 1 mM IPTG to a final concentration for 3 h. Cells were then centrifuged at 7,500 × *g* for 30 min at 4℃. The pellet was subsequently resuspended in buffer A20 (20 mM HEPES pH 7.9, 100 mM KCl, 5 mM $MgCl_2$, 10% glycerol, and 20 mM imidazole) plus protease inhibitor (Complete Ultra Tablets, Mini, EDTA-free, EASYpack, Roche). The cells were then disrupted by sonication, and the soluble fraction was collected after centrifugation at 12,000 × *g* for 30 min at 4℃. The supernatant was used for protein purification by immobilized-metal affinity chromatography (IMAC) using HisPur Ni-NTA Superflow Agarose (*Thermo Scientific*). The recombinant IrmA protein was eluted from Ni-NTA resin by increasing the concentration of imidazole using buffer A200 (20 mM HEPES pH 7.9, 100 mM KCl, 5 mM $MgCl_2$, 10% glycerol, and 200 mM imidazole). The purified His-tagged IrmA protein was adjusted to 1 mg/mL and inoculated into rabbits according to standard protocols (Unitatd'Experimentació Animal de Farmàcia–CCiTUB. Universitat de Barcelona, Barcelona, Spain). After immunization, preimmune serum and serum collected after the immunization period were tested against the IrmA protein by Western blotting.

**Electrophoresis and Western blotting of proteins.** Whole-cell protein extracts were prepared in Laemmli buffer (glycerol 5% vol/vol, $\beta$-mercaptoethanol 2.5%, SDS 1.15% p/v, Tris–HCl 31 mM pH 6.6 and bromophenol blue 0.05%). Protein samples were analyzed by Tris-Glycine-SDS triphasic gels with 16.5%

polyacrylamide. Proteins were transferred from the gels to PVDF membranes by using a semidry electrophoretic transfer cell (Bio–Rad) at 15 V for 40 min. For Western blot analysis, a monoclonal antibody directed against the Flag epitope (Sigma–Aldrich) or a polyclonal antibody directed against the IrmA protein were diluted 1:10.000 or 1:1000 in a solution of PBS, 0.2% Triton and 3% skimmed milk. The membranes containing the proteins were incubated with the diluted antibody for 16 h at 4°C. The membranes were then washed for 10 min with PBS and 0.2% Triton X-100 (Sigma–Aldrich) solution. The washing step was repeated three times. Thereafter, the membranes were incubated with horseradish peroxidase-conjugated goat anti-mouse IgG (Promega) or anti-rabbit IgG (Promega) diluted 1:2.500 in a solution of PBS and 0.2% Triton X-100 for 45 min at room temperature. Again, three washing steps with PBS and 0.2% Triton solution were performed for 30 min each, and immunodetection of the specific protein was performed by enhanced chemiluminescence by using the Molecular Imager ChemiDoc XRS system and Quantity One software (Bio–Rad).

**Isolation of RNA.** For RNA isolation, bacterial cells were grown to an $OD_{600nm}$ of 2.0 at 37°C. Then, 5 mL of cells was mixed with a 0.2 × volume of a stop solution (95% ethanol, 5% phenol), shaken and centrifuged for 10 min at 6,000 × $g$. Bacterial pellets were subsequently frozen at −80°C until use. Total RNA was extracted from the bacterial pellets by using Tripure Isolation Reagent (Roche) according to the manufacturer's instructions. Potential traces of DNA were removed by digestion with Turbo DNA-free (Thermo Scientific) according to the manufacturer's instructions. RNA concentration and quality were measured using a NanoDrop 1000 (Thermo Scientific).

**Walking RT–PCR.** The walking RT–PCR assay was performed following the methodology previously described in (47).

**5′ RACE.** 5′ RACE experiments were performed as previously described in (48).

**Bioinformatics analysis.** The *flu-yeeR-irmA* alignment was generated using Easyfig (49). The evolutionary history was inferred using the UPGMA method (50). The optimal tree with a total branch length equal to 0.03874617 is shown. The tree is drawn to scale, with branch lengths in the same units as those of the evolutionary distances used to infer the phylogenetic tree. The evolutionary distances were computed using the maximum composite likelihood method (51) and are in units of the number of base substitutions per site. The analysis involved 14 nucleotide sequences. The codon positions included were 1st + 2nd + 3rd+Noncoding. All positions containing gaps and missing data were eliminated. There was a total of 956 positions in the final data set. Evolutionary analyses were conducted in MEGA7 (52).

The prediction analysis of the *irmA* promoter was performed using the BPROM software (https://www.softberry.com/berry.phtml?topic=bprom&group=programs&subgroup=gfindb).

## SUPPLEMENTAL MATERIAL

Supplemental material is available online only.

**SUPPLEMENTAL FILE 1**, PDF file, 1.6 MB.

## ACKNOWLEDGMENTS

This work was supported by grants from Fundació "La Marató TV3," Spain (project 201818 10), and BIO2016-76412-C2-1-R and PID2019-107479RB-I00 (AEI/FEDER, UE) from the Ministerio de Economía, Industria y Competitividad, and CERCA Program/Generalitat de Catalunya to A.J. M.B. was the recipient of an FI fellowship from the Generalitat de Catalunya. The funders had no role in data collection, interpretation, study design, or the decision to publish these data.

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
