## [Reviewer comments · Microbiology Spectrum]

Microbiology Spectrum

Differential expression of two copies of the *irmA* gene in the enteroaggregative *E. coli* strain 042.

Manuel Bernabeu, Sonia Aznar, Alejandro Prieto, Mário Hüttener, and Antonio Juarez

Corresponding Author(s): Antonio Juarez, UNIVERSITAT DE BARCELONA

Review Timeline:

Submission Date:	February 4, 2022
Editorial Decision:	March 18, 2022
Revision Received:	June 6, 2022
Editorial Decision:	June 7, 2022
Revision Received:	June 9, 2022
Accepted:	June 9, 2022

Editor: Sandeep Tamber

Reviewer(s): The reviewers have opted to remain anonymous.

Transaction Report:

DOI: <https://doi.org/10.1128/spectrum.00454-22>

March 18, 2022

Prof. Antonio Juárez
University of Barcelona
Department of Microbiology
Barcelona
Spain

Re: Spectrum00454-22 (Differential expression of two copies of the *irmA* gene in the enteroaggregative *E. coli* strain 042.)

Dear Prof. Antonio Juárez:

Link Not Available

Sincerely,

Sandeep Tamber

Journals Department
Reviewer comments:

Reviewer #1 (Comments for the Author):

This manuscript investigates the biological role of the two copies of the *irmA* gene in the enteroaggregative *E. coli* strain 042. The two *irmA* alleles, *irmA*_2244 and *irmA*_4509, exhibit differences in expression and regulation, and the authors provide experimental data to support this interpretation. The authors conclude that the *irmA*_2244 allele plays a backup role to ensure *IrmA* expression when the *irmA*_4509 allele loses its function.

Overall comment

The authors make the observation that gene duplication has occurred, leading to 2 copies of *irmA*. Importantly, the *irmA* gene is located within a pathogenicity island (PAI), which itself is mobile. Therefore, I would posit that the most likely mechanism of duplication is that the *irmA* gene (and its associated PAI) was acquired in 2 independent PAI acquisition events, leading to its duplication on the chromosome. How much of the region surrounding *irmA* is duplicated in 042? Are the 2 *irmA* alleles located on different PAIs? These questions should be investigated and discussed.

Specific comments

L95-95. The text needs correction. The IrmA protein was not detected in the double *irmA_2244 irmA_4509* mutant derivative.

Fig 1. Please add relevant molecular weight markers and indicate position of IrmA. Panel B is not needed. I suggest combining this Figure with Fig 2.

Fig 2. Panel B is not needed.

L124. g/l (not gr/l)

Fig 3. Panels B, E, H are not needed.

L131-141. Are either of the copies of the *hha* gene located near the *IrmA* genes on the chromosome?

Fig 4. Panel C is not needed.

Fig 5. Panels C and D are not needed. Fig 5 and Fig 6 can be combined.

Fig 6. Panel B is not needed.

Fig 7. Panel B is not needed. Fig 7 and Fig 8 can be combined.

Fig 8. Panels C and D are not needed.

Fig 11. Based on the position of the TSS, the positions of the -10 and -35 boxes are incorrectly noted. The true -10 box is a very weak consensus sequence. The 'A' nucleotide in *irmA_4509* upstream of the +1 site (currently labelled as position 13) is likely the first 'T' in the TATAAT box, and could explain the differences in transcription between the 2 alleles. Please correct the figure and discuss this possibility.

L202-219. Based on the above data, the authors should change the 'A' nucleotide to 'G' in their pUJ8 promoter probe vector and test this.

L202-219. The 3,4,13,23 nucleotides should be labelled according to convention based on their position relative to the TSS.

L202-219. How was the TSS of *irmA_2244* determined given transcription is undetectable?

Fig 14. Panel D is not needed.

L220-227. I can't see how this section is informative. Essentially, it just shows *irmA* is located between *flu* and *yeeR*, which is already established.

L279. Either show the data or remove the comment. There is no place for unpublished data in the literature.

Reviewer #3 (Comments for the Author):

This study analyzes the expression of two interleukin receptor mimic protein A (IrmA)-encoding alleles in enteroaggregative *E. coli* strain 042. With the help of C-terminal Flag-tag fusions and with transcriptional *lacZ* fusions of both alleles, *irmA_2244* and *irmA_4509*, the authors were able to demonstrate that expression of *irmA_4509* was affected by growth temperature (higher expression at 37 {degree sign}C than at 25 {degree sign}C), nutrient availability (higher expression in minimal medium than in LB), and oxygen availability (higher expression in the presence of oxygen than under anaerobic conditions). Deletion of *hha* or the second *hha* allele present in EAEC strain 042 led to significant upregulation of *irmA_4509* expression. Whereas oxidative stress (due to incubation with hydrogen peroxide) did not induce *irmA_4509* expression, the deletion of *oxyR* resulted in a significantly increased *irmA_4509* expression. Under all these conditions, expression of *irmA_2244* was almost not detectable and not affected. An exception was the *irmA_4509 oxyR* double mutant, in which a significantly increased expression of *irmA_2244* could then be shown.

The authors provide experimental evidence that in *E. coli* 042 expression of *irmA* alleles is, in contrast to UPEC strain CFT073,

directed from a promoter located in the yeeR-irmA intergenic region. They mapped the promoter by 5'-RACE and showed that single nucleotide differences in the promoter sequences of the two irmA alleles are responsible for their different expression profile.

Finally, the authors bioinformatically analyzed the flu-yeeR-irmA gene cluster in ten selected E. coli strains representing different pathotypes. They observed that this regions is structurally instable with duplications, deletions, and insertions occurring between flu and yeeR. Four out of the ten strains analyzed displayed duplications of the flu-yeeR-irmA region. Nucleotide sequence comparison of the yeeR-irmA intergenic regions indicated that the irmA_4509 upstream region is closely related to the upstream regions of the irmA alleles in EAEC strains 55989 and 0104:H4, whereas the upstream region of irmA_2244 is rather distantly related to that of irmA_4509.

The authors discuss their findings in the context of different gene duplication scenarios (gene loss and fixation, genetic backup). This is an interesting study with regard to the consequences of gene duplications and the role of irmA in EAEC.

Important points for improvement of the study:

1) The authors show in Figure 14 that an irmA:lacZ transcriptional fusion with the mutated yeeR-irmA_4509 intergenic regions behaved like the transcriptional fusion based on the yeeR-irmA_2244 intergenic region. They indicated four nucleotide residues which they have mutated. Unfortunately, these four nucleotide residues have not been tested individually for their effect on the transcriptional activity. Therefore it is difficult to assess which contribution the individual point mutations have on the promoter activity of the two irmA alleles. There is another nucleotide difference (upstream of the -35 region) between the upstream region of both alleles. Why didn't the authors include this SNP as well in their analysis? It will be important to understand the impact of these point mutations to see whether simply the -10 region is impaired or not. Accordingly, the authors should repeat the reporter gene assays with corresponding transcriptional fusions which specifically address the role of the individual point mutations in the yeeR-irmA intergenic region.

2) The authors write (l. 196-199) that they mutated the flu4511 promoter and then compared IrmA expression in the strains 042 wt, 042 irmA_4509xFLAG and 042 irmA_4509xFLAG p4511. As a control, it would be important to show that mutation of the flu4511 promoter indeed led to a reduced expression of antigen 43.

3) The grouping of irmA upstream regions as shown in the phylogenetic tree (Figure 15b) is interesting and will be an important basis for further functional analyses to understand the sequence diversity of the irmA upstream regions. The role of interleukin receptor mimic proteins is so far unclear. The authors interpretation that gene duplication resulting in differentially regulated allelic variants may be advantageous under different growth conditions sounds good. Do irmA allelic variants with highly similar irmA upstream regions exhibit the same irmA expression profile? Does, for example, the irmA allele of the O26 H11 strain show the same expression characteristics as irmA_2244 of EAEC strain 042? What about the two irmA alleles present in H10407, or the allele 5250 of the O145 H28? This would be very interesting to understand the impact of nucleotide sequence variability in the irmA upstream region for gene expression.

4) Can the authors comment on the role of horizontal gene transfer (HGT) in the "gene duplication" events relevant for irmA? The authors write that the entire flu-yeeR-irmA region can be duplicated. This regions belongs to a pathogenicity island. If the authors would extend their phylogenetic analysis further to a larger sequence context up- and downstream of irmA, they may be able to see whether the relatedness of the yeeR-irmA intergenic regions correlates with the relatedness of a larger sequence neighbourhood, which may indicate acquisition of these genes by HGT and homologous recombination.

5) What still remains unknown is whether both irmA allelic variants are indeed functional and whether both are expressed in vivo or during infection. Do both mutants of EAEC strain 042 differ in their growth or survival behaviour? Are any fitness defects observable?

Staff Comments:

Preparing Revision Guidelines

- Point-by-point responses to the issues raised by the reviewers in a file named "Response to Reviewers," NOT IN YOUR COVER LETTER.
- Upload a compare copy of the manuscript (without figures) as a "Marked-Up Manuscript" file.
- Each figure must be uploaded as a separate file, and any multipanel figures must be assembled into one file.
- Manuscript: A .DOC version of the revised manuscript

- Figures: Editable, high-resolution, individual figure files are required at revision, TIFF or EPS files are preferred

Please return the manuscript within 60 days; if you cannot complete the modification within this time period, please contact me. If you do not wish to modify the manuscript and prefer to submit it to another journal, please notify me of your decision immediately so that the manuscript may be formally withdrawn from consideration by Microbiology Spectrum.

Reviewer #1 (Comments for the Author):

This manuscript investigates the biological role of the two copies of the *irmA* gene in the enteroaggregative *E. coli* strain 042. The two *irmA* alleles, *irmA*_2244 and *irmA*_4509, exhibit differences in expression and regulation, and the authors provide experimental data to support this interpretation. The authors conclude that the *irmA*_2244 allele plays a backup role to ensure *IrmA* expression when the *irmA*_4509 allele loses its function.

Overall comment

The authors make the observation that gene duplication has occurred, leading to 2 copies of *irmA*. Importantly, the *irmA* gene is located within a pathogenicity island (PAI), which itself is mobile. Therefore, I would posit that the most likely mechanism of duplication is that the *irmA* gene (and its associated PAI) was acquired in 2 independent PAI acquisition events, leading to its duplication on the chromosome. How much of the region surrounding *irmA* is duplicated in 042? Are the 2 *irmA* alleles located on different PAIs? These questions should be investigated and discussed.

We already showed that the *irmA* alleles are located in two large regions that are duplicated in strain 042 (see Bernabeu et al BMC Genomics. 2019 Apr 24;20(1):313). Hence, they do not have been independently acquired. Comments have been introduced in the discussion section

Specific comments

L95-95. The text needs correction. The *IrmA* protein was not detected in the double *irmA*_2244 *irmA*_4509 mutant derivative.

Corrected

Fig 1. Please add relevant molecular weight markers and indicate position of *IrmA*. Panel B is not needed. I suggest combining this Figure with Fig 2.

Done

Fig 2. Panel B is not needed.

Panels B from Figs 1 and 2 have been deleted, and Figs. 1 and 2 have been combined

L124. g/l (not gr/l)

Corrected

Fig 3. Panels B, E, H are not needed.

They have been suppressed

L131-141. Are either of the copies of the *hha* gene located near the *IrmA* genes on the chromosome?

Comments have been introduced in the text (results section)

Fig 4. Panel C is not needed.

Suppressed

Fig 5. Panels C and D are not needed. Fig 5 and Fig 6 can be combined.

Combined

Fig 6. Panel B is not needed.

Suppressed

Fig 7. Panel B is not needed. Fig 7 and Fig 8 can be combined.

Combined

Fig 8. Panels C and D are not needed.

Suppressed

Fig 11. Based on the position of the TSS, the positions of the -10 and -35 boxes are incorrectly noted. The true -10 box is a very weak consensus sequence. The 'A' nucleotide in *irmA*_4509 upstream of the +1 site (currently labelled as position 13) is likely the first 'T' in the TATAAT box, and could explain the differences in transcription between the 2 alleles. Please correct the figure and discuss this possibility. The reviewer is right. We already noticed this, but decided to show the results of the bioinformatic analysis. Our aim was not to precisely identify the -35 and -10 promoter regions, but to show nucleotide differences in both promoter regions would account for the observed differences in expression. Throughout the manuscript, we do not refer to the "*irmA* promoter", but to the "*irmA* promoter region", and to the "bioinformatically identified putative promoter". Comments have been introduced in the results section.

L202-219. Based on the above data, the authors should change the 'A' nucleotide to 'G' in their pUJ8 promoter probe vector and test this.

When, instead of performing multiple changes in the promoter region, as reported, performed single changes, differences in expression were not significant

L202-219. The 3,4,13,23 nucleotides should be labelled according to convention based on their position relative to the TSS.

Done

L202-219. How was the TSS of *irmA*_2244 determined given transcription is undetectable?

Transcription is undetectable by using gene fusions in the wt system, but we coded the regions containing the putative promoters in the pUJ8 multicopy plasmid, and a low transcriptional activity for the *irmA*_2244 allele could be detected. Both RACE and walking PCR highly amplify the low levels of transcription of that gene.

Fig 14. Panel D is not needed.

Supressed

L220-227. I can't see how this section is informative. Essentially, it just shows *irmA* is located between *flu* and *yeeR*, which is already established.

The analysis performed shows the presence of duplications of that region in several strains, as well as the differential presence of insertions and deletions.

L279. Either show the data or remove the comment. There is no place for unpublished data in the literature.

Comment removed

Reviewer #3 (Comments for the Author):

This study analyzes the expression of two interleukin receptor mimic protein A (*IrmA*)-encoding alleles in enteroaggregative *E. coli* strain 042. With the help of C-terminal Flag-tag fusions and with transcriptional *lacZ* fusions of both alleles, *irmA_2244* and *irmA_4509*, the authors were able to demonstrate that expression of *irmA_4509* was affected by growth temperature (higher expression at 37 (degree sign)C than at 25 (degree sign)C), nutrient availability (higher expression in mimimal medium than in LB), and oxygen availability (higher expression in the presence of oxygen than under anaerobic conditions). Deletion of *hha* or the second *hha* allele present in EAEC strain 042 led to significant upregulation of *irmA_4509* expression. Whereas oxidative stress (due to incubation with hydrogen peroxide) did not induce *irmA_4509* expression, the deletion of *oxyR* resulted in a significantly increased *irmA_4509* expression. Under all these conditions, expression of *irmA_2244* was almpst not detectable and not affected. An exception was the *irmA_4509 oxyR* double mutant, in which a significantly increased expression of *irmA_2244* could then be shown.

The authors provide experimental evidence that in *E. coli* 042 expression of *irmA* alleles is, in contrast to UPEC strain CFT073, directed from a promoter located in the *yeeR-irmA* intergenic region. They mapped the promoter by 5'-RACE and showed that single nucleotide differences in the promoter sequences of the two *irmA* alleles are responsible for their different expression profile.

Finally, the authors bioinformatically analyzed the *flu-yeeR-irmA* gene cluster in ten selected *E. coli* strains representing different pathotypes. They observed that this regions is structurally instable with duplications, deletions, and insertions occurring between *flu* and *yeeR*. Four out of the ten strains analyzed displayed duplications of the *flu-yeeR-irmA* region. Nucleotide sequence comparison of the *yeeR-irmA* intergenic regions indicated that the *irmA_4509* upstream region is closely related to the upstream regions of the *irmA* alleles in EAEC strains 55989 and 0104:H4, whereas the upstream region of *irmA_2244* is rather distantly related to that of *irmA_4509*.

The authors discuss their findings in the context of different gene duplication scenarios (gene loss and fixation, genetic backup). This is an interesting study with regard to the consequences of gene duplications and the role of *irmA* in EAEC.

Important points for improvement of the study:

1) The authors show in Figure 14 that an *irmA:lacZ* transcriptional fusion with the mutated *yeeR-irmA_4509* intergenic regions behaved like the transcriptional fusion based on the *yeeR-irmA_2244* intergenic region. They indicated four nucleotide residues which they have mutated. Unfortunately, these four nucleotide residues have not been tested individually for their effect on the transcriptional activity. Therefore it is difficult to assess which contribution the individual point mutations have on the promoter activity of the two *irmA* alleles. There is another nucleotide difference (upstream of the -35 region) between the upstream region of both alleles. Why didn't the authors include this SNP as well in their analysis? It will be important to understand the impact of these point mutations to see whether simply the -10 region is impaired or not. Accordingly, the authors should repeat the reporter gene assays with corresponding transcriptional fusions which specifically address the role of the individual point mutations in the *yeeR-irmA* intergenic region.

Our aim was to show that *irmA* transcripts start at the intergenic *yeeR irmA* region and not downstream the *flu* promoter, and that some of the observed differences in that region account for the differential expression of both *irmA* alleles, and this is clearly shown. When individual point mutations are tested, the effect on *irmA* expression is low and not significant. It is apparent that the combination of different point mutations accounts for the observed differences in the expression of both alleles.

2) The authors write (l. 196-199) that they mutated the *flu4511* promoter and then compared *IrmA* expression in the strains 042 wt, 042 *irmA_4509xFLAG* and 042 *irmA_4509xFLAG p4511*. As a control, it would be important to show that mutation of the *flu4511* promoter indeed led to a reduced expression of antigen 43.

The *flu* promoter has been well-characterized in different reports, and alterations in its sequence must necessarily alter *flu* transcription. In any case, we found that the experiment suggested by the reviewer had to be made. A 315 bp including the *flu* promoter was deleted in the 042 strain. Transcription of the *flu 4511* allele both the wt 042 strain and its derivative lacking the *flu* promoter was determined by isolating RNA, followed by its retrotranscription to cDNA. Thereafter, both 16S RNA and the Agn43 4511 transcript were amplified with specific primers (see Figure). As expected, transcription of the Agn43 4511 gene was significantly reduced, although not completely abolished. This is likely due to the fact that the primers used to amplify the transcripts corresponding to the 4511 allele likely amplify transcripts corresponding to the 2242 allele.

3) The grouping of *irmA* upstream regions as shown in the phylogenetic tree (Figure 15b) is interesting and will be an important basis for further functional analyses to understand the sequence diversity of the *irmA* upstream regions. The role of interleukin receptor mimic proteins is so far unclear. The authors interpretation that gene duplication resulting in differentially regulated allelic variants may be advantageous under different growth conditions sounds good. Do *irmA* allelic variants with highly similar *irmA* upstream regions exhibit the same *irmA* expression profile? Does, for example, the *irmA* allele of the O26 H11 strain show the same expression characteristics as *irmA*_2244 of EAEC strain 042? What about the two *irmA* alleles present in H10407, or the allele 5250 of the O145 H28? This would be very interesting to understand the impact of nucleotide sequence variability in the *irmA* upstream region for gene expression.

Some of these strains are not available at our lab, and, whereas interesting, these studies are out of the scope of our work.

4) Can the authors comment on the role of horizontal gene transfer (HGT) in the "gene duplication" events relevant for *irmA*? The authors write that the entire *flu-yeeR-irmA* region can be duplicated. This region belongs to a pathogenicity island. If the authors would extend their phylogenetic analysis further to a larger sequence context up- and downstream of *irmA*, they may be able to see whether the relatedness of the *yeeR-irmA* intergenic regions correlates with the relatedness of a larger sequence neighbourhood, which may indicate acquisition of these genes by HGT and homologous recombination. **We already reported on the characterization of that duplicated region in strain 042 (see (see Bernabeu et al BMC Genomics. 2019 Apr 24;20(1):313).**

5) What still remains unknown is whether both *irmA* allelic variants are indeed functional and whether both are expressed *in vivo* or during infection. Do both mutants of EAEC strain 042 differ in their growth or survival behaviour? Are any fitness defects observable?

We checked both mutants and the wt strain for growth differences at different temperatures and in different *in vitro* growth conditions, and no significant differences were found

June 7, 2022

Prof. Antonio Juarez
UNIVERSITAT DE BARCELONA
Barcelona
Spain

Re: Spectrum00454-22R1 (Differential expression of two copies of the *irmA* gene in the enteroaggregative *E. coli* strain 042.)

Dear Prof. Antonio Juarez:

Thank you for submitting your manuscript to Microbiology Spectrum. As you will see your paper is very close to acceptance. Please modify the manuscript along the lines I have recommended. As these revisions are quite minor, I expect that you should be able to turn in the revised paper in less than 30 days, if not sooner. If your manuscript was reviewed, you will find the comments below.

When submitting the revised version of your paper, please provide (1) point-by-point responses to the issues I raised in your cover letter, and (2) a PDF file that indicates the changes from the original submission (by highlighting or underlining the changes) as file type "Marked Up Manuscript - For Review Only". Please use this link to submit your revised manuscript. Detailed instructions on submitting your revised paper are below.

Link Not Available

Sincerely,

Sandeep Tamber

Editor comments:

Further to reviewer #3 point 1: Please indicate in the final manuscript text that the individual point mutations did not have a significant effect on *irmA* expression (data not shown, or show in the supplement). Highlight this new sentence in the marked up manuscript.

Further to reviewer #3 point 2: Please include the new data showing the effect of the flu promotor on Ag 43 expression in the supplementary. Make note of this control in the final manuscript. Highlight the change in the marked up manuscript.

Preparing Revision Guidelines

- point-by-point responses to the issues I raised in your cover letter
- Upload a compare copy of the manuscript (without figures) as a "Marked-Up Manuscript" file.
- Each figure must be uploaded as a separate file, and any multipanel figures must be assembled into one file.
- Manuscript: A .DOC version of the revised manuscript
- Figures: Editable, high-resolution, individual figure files are required at revision, TIFF or EPS files are preferred

Please return the manuscript within 60 days; if you cannot complete the modification within this time period, please contact me. If you do not wish to modify the manuscript and prefer to submit it to another journal, please notify me of your decision immediately so that the manuscript may be formally withdrawn from consideration by Microbiology Spectrum.

Following Editor's suggestions, (i) a sentence indicating that the individual point mutations at the 042 *irmA_4509* promoter region did not have a significant effect on IrmA expression has been introduced at the end of the results subsection **"Alterations in the *irmA* promoter regions account for the differential expression of the *irmA_2244::lacZ* and *irmA_4509::lacZ* alleles"**.(ii) a sentence indicating that "deletion of the *flu p4511* promoter did not influence IrmA expression (Figure 9), but significantly reduced *flu* transcript levels (Figure S4)" has been introduced at the end of the results subsection **"*irmA* transcripts are initiated downstream of the *yeeR* gene in strain 042"**. (iii) Accordingly, the S4 supplementary figure showing the effect of the deletion of the *flu p4511* promoter on the transcription of the *flu 4511* allele and its corresponding legend have been introduced in the Supplementary information file.

June 9, 2022

Prof. Antonio Juarez
UNIVERSITAT DE BARCELONA
Barcelona
Spain

Re: Spectrum00454-22R2 (Differential expression of two copies of the *irmA* gene in the enteroaggregative *E. coli* strain 042.)

Dear Prof. Antonio Juarez:

Your manuscript has been accepted, and I am forwarding it to the ASM Journals Department for publication. You will be notified when your proofs are ready to be viewed.

Sincerely,

Sandeep Tamber
Editor, Microbiology Spectrum
